# C-phycocyanin from *Limnothrix* Species KNUA002 Alleviates Cisplatin-Induced Ototoxicity by Blocking the Mitochondrial Apoptotic Pathway in Auditory Cells

**DOI:** 10.3390/md17040235

**Published:** 2019-04-19

**Authors:** Ye-Ri Kim, Jeong-Mi Do, Kyung Hee Kim, Alexandra R. Stoica, Seung-Woo Jo, Un-Kyung Kim, Ho-Sung Yoon

**Affiliations:** 1Department of Biology, College of Natural Sciences, Kyungpook National University, Daegu 41566, Korea; yell_90@knu.ac.kr (Y.-R.K.); leciel631@naver.com (J.-M.D.); kkh5566@knu.ac.kr (K.H.K.); alexa_stoica93@yahoo.ro (A.R.S.); jsw8796@gmail.com (S.-W.J.); 2School of Life Sciences, BK21 Plus KNU Creative BioResearch Group, Kyungpook National University, Daegu 41566, Korea; 3Advanced Bio-Resource Research Center, Kyungpook National University, Daegu 41566, Korea

**Keywords:** ototoxicity, cisplatin, C-phycocyanin, *Limnothrix*, HEI-OC1

## Abstract

Ototoxicity, or adverse pharmacological effects on the inner ear or auditory nerve, is a common side effect of cisplatin, a platinum-based drug widely used in anticancer chemotherapy. Although the incidence of ototoxicity is high among patients that receive cisplatin therapy, there is currently no effective treatment for it. The generation of excessive reactive oxygen species (ROS) is considered to be the major cause of cisplatin-induced ototoxicity. C-phycocyanin (C-PC), a blue phycobiliprotein found in cyanobacteria and red algae, has antioxidant and anticancer activities in different experimental models in vitro and in vivo. Thus, we tested the ability of C-PC from *Limnothrix* sp. KNUA002 to protect auditory cells from cisplatin-induced ototoxicity in vitro. Pretreatment with C-PC from *Limnothrix* sp. KNUA002 inhibited apoptosis and protected mitochondrial function by preventing ROS accumulation in cisplatin-treated House Ear Institute-Organ of Corti 1 (HEI-OC1) cells, a mouse auditory cell line. Cisplatin increased the expression of Bax and reduced the expression of Bcl-2, which activate and inhibit, respectively, the mitochondrial apoptotic pathway in response to oxidative stress. Pretreatment with C-PC prior to cisplatin treatment caused the Bax and Bcl-2 levels to stay close to the levels in untreated control cells. Our results suggest that C-PC from *Limnothrix* sp. KNUA002 protects cells against cisplatin-induced cytotoxicity by inhibiting the mitochondrial apoptotic pathway.

## 1. Introduction

Phycobiliproteins (PBPs) are major components of the phycobilisome, the light-harvesting antenna for photosystem II in cyanobacteria and some red algae. PBPs can be divided on the basis of their spectral properties into three major groups: phycoerythrins, phycocyanins, and allophycocyanins. Each PBP is composed of a heterodimer containing equal numbers of α-subunits and β-subunits covalently bound by one or two cysteine thioether linkages to an open-chain tetrapyrrole chromophore, known as phycobilin. Phycobilin is structurally similar to the bile pigment biliverdin, which can be converted to bilirubin, an endogenous antioxidant, by biliverdin reductase.

C-phycocyanin (C-PC) is a blue PBP found in cyanobacteria and red algae. C-PC accounts for up to 20% of all the protein in cyanobacteria and is used in many commercial applications as a natural colorant in food or cosmetics, as a fluorescent dye, and as a nutraceutical supplement in biomedical research [1]. There is increasing interest in potential pharmaceutical uses of C-PC that take advantage of its antioxidant, anti-inflammatory, hepatoprotective, and anticancer effects [2,3].

Ototoxicity is the cellular degeneration of cochlear tissues due to the use of certain therapeutic agents [4]. Platinum-based chemotherapeutic agents, aminoglycoside antibiotics, loop diuretics, macrolide antibiotics, and antimalarials all have ototoxic effects [4]. Cisplatin (cis-diamminedichloroplatinum II) and carboplatin (cis-diammine 1,1-cyclobutane dicarboxylatoplatinum II) [5] are commonly used platinum-based anticancer drugs. Cisplatin-induced ototoxicity is usually progressive, dose-dependent, bilateral, and irreversible, resulting in sensorineural hearing loss. Individuals with cancers requiring irradiation of the base of the skull or brain are particularly vulnerable to cisplatin-induced ototoxicity [6], although the underlying molecular mechanism of cisplatin-induced ototoxicity is not fully understood. The generation of excessive ROS is considered to be one of the causative pathogenesis of cisplatin-induced ototoxicity [7]. In the antioxidant model, excessive ROS formation within the cochlea leads to a reduction in antioxidant enzymes following exposure to cisplatin chemotherapy [8,9].

C-PC prevents apoptosis in mice with cisplatin-induced nephrotoxicity [10] by preventing oxidative stress and preserving the activity of antioxidant enzymes [11]. It is not yet known if C-PC has similar effects on other side effects caused by cisplatin. In this study, we show that C-PC from *Limnothrix* sp. KNUA002 has protective effects against cisplatin-induced ROS accumulation and ototoxicity in the mouse auditory cell line HEI-OC1.

## 2. Results and Discussion

### 2.1. C-PC Alleviates Cisplatin-Induced Apoptosis in HEI-OC1 Cells

We checked for cytotoxicity in cells treated with 0.1, 0.5, 1, 2, 5, 10, or 20 μg/mL C-PC alone before we assessed the protective efficacy of C-PC in cisplatin-treated cells. C-PC did not cause any cytotoxicity up to a concentration of 5 μg/mL, although it did reduce cell viability at higher concentrations (Figure 1A). Next, we measured the cell viability after cisplatin treatment with or without pretreatment with 0.1–20 μg/mL C-PC. Treatment with cisplatin alone reduced the cell viability to around 40% of that of untreated control cells, while pretreatment with 1 μg/mL or 2 μg/mL C-PC rescued the cell viability after cisplatin treatment to approximately 62% of that of the control cells (Figure 1B).

To determine if C-PC provides protection against cisplatin-induced cell death, we determined the cell cycle phase and the activation of the apoptotic pathway in cisplatin-treated cells. Cisplatin arrests the cell cycle by entering the nucleus and binding with DNA. The proportion of cells in the sub-G0/G1 phase of the cell cycle increased after cisplatin treatment (Figure 2A). About 15% of the cells were in the sub-G0/G1 phase after cisplatin treatment alone; however, pretreatment with 1 μg/mL C-PC reduced the sub-G0/G1 fraction after cisplatin treatment to about 7% (Figure 2B). Those results indicate that C-PC defends the cell from the cell cycle arrest caused by cisplatin. DNA fragmentation occurs during apoptosis, exposing the N terminus of the DNA molecules and allowing fluorescent substances to bind to the fragmented ends. Expression of caspase-3 followed a similar pattern—cisplatin treatment caused a massive increase in caspase-3 expression, which was almost completely reversed by pretreatment with C-PC (Figure 2C). In a terminal deoxynucleotidyl transferase dUTP nick-end labeling (TUNEL) assay, cisplatin treatment alone resulted in a strong fluorescent signal, which was substantially weakened by pretreatment with C-PC (Figure 2D). The results of the caspase-3 assay and the TUNEL assay showed that C-PC has a capacity to protect HEI-OC1 cells from cisplatin-induced apoptosis.

We found that C-PC from *Limnothrix* sp. KNUA002 protects cells of HEI-OC1 against cisplatin-induced ototoxicity by reducing ROS-induced damage to the mitochondria. Two of the most widely studied effects of C-PC in vitro and in vivo are its antioxidant capacity and its free radical scavenging ability [11]. In particular, Spirulina species, a rich natural source of phycocyanin, have been studied as a nutraceutical in health foods [12]. Bermejo-Bescós et al. reported that protean extract and phycocyanin of *Spirulina platensis* increased antioxidant enzymatic activities and ROS scavenging in SH-SY5Y neuroblastoma cells and protected those cells against oxidative stress caused by iron deficiency [13]. Abd El-Baky et al. demonstrated that endogenous antioxidant enzymatic activity in *S. platensis* cells increased with increasing hydrogen peroxide concentration [14]. Similar to the observations made in in vitro studies, Spirulina increases antioxidant enzymatic activities, reduces lipid peroxidation and DNA damage, and scavenges free radicals in several animal models [12]. C-PC from Spirulina prevented cisplatin-induced nephrotoxicity by attenuating oxidative damage and preserving the activity of antioxidant enzymes in the kidneys of cisplatin-treated rodents [11,15].

There have been a number of in-depth studies of the potential for Spirulina to be used against various diseases; however, few studies have evaluated the application of C-PC from other species [16]. *Limnothrix* sp. produced C-PC at a level similar to that produced by Spirulina and had antioxidant activity on DPPH (2,2-Diphenyl-1-picrylhydrazyl) free radicals [17]. Our present results demonstrate that C-PC from *Limnothrix* sp. KNUA002 decreases oxidant stress on cells damaged by cisplatin in vitro, but further study is required to determine if C-PC as a nutraceutical supplement will be effective against cisplatin-induced ototoxicity in mice.

### 2.2. C-PC Effectively Reduces Intracellular ROS Levels in HEI-OC1 Cells

We investigated the effects of C-PC on the intracellular ROS level, which is the main determinant of cisplatin-induced apoptosis. In fluorescent dye 2′, 7′-dichlorodihydrofluorescein diacetate (DCFH-DA) assays, cisplatin treatment alone increased the relative fluorescence-activated cell sorting (FACS) fluorescence intensity by approximately 30% relative to that in control cells, while pretreatment with C-PC decreased the fluorescence intensity by 20% relative to that in the cells treated with cisplatin alone (Figure 3A,B). Those results suggest that accumulation of intracellular ROS played a role in the cisplatin-induced apoptosis, and that C-PC protected the cells from ROS accumulation.

### 2.3. C-PC Regulates the Changes in Bax and Bcl-2 Protein Levels Caused by Cisplatin-induced ROS Accumulation

The main apoptosis-related proteins associated with ROS in mitochondria are the pro-apoptotic Bax and the anti-apoptotic Bcl-2. Cisplatin treatment alone increased Bax expression and decreased Bcl-2 expression in HEI-OC1 cells compared with that in control cells, while pretreatment with C-PC kept the expression levels of both proteins close to those in the control cells (Figure 4A,B). Bax is present in the cytoplasm and induces apoptosis by migrating into the mitochondrial intermembrane space and causing the release of cytochrome c in response to oxidative stress. Bcl-2 restrains apoptosis and facilitates cell survival by inhibiting Bax activity. The accumulation of intracellular ROS induced by cisplatin activates Bax and causes apoptosis by depriving the cell of mitochondrial function. C-PC protects the cell by reducing the ROS level, thus suppressing the Bax pathway (Figure 4C).

There is increasing evidence that mitochondrial ROS production plays a key role in hearing loss by activating the apoptotic pathway in cochlear hair cells [18]. Bcl-2, an upstream regulator of caspase signaling, is localized to the outer mitochondrial membrane and acts by inhibiting cytochrome c release from the mitochondria, thus preventing activation of caspase and the apoptotic pathway. Excess ROS accumulation following excessive noise or exposure to ototoxic drugs leads to downregulation of Bcl-2 [19]. Therefore, researchers hypothesized that oxidative stress induces activation of the apoptotic pathway in cochlear hair cells by blocking Bcl-2 activation [20]. Consistent with that hypothesis, our results showed that pre-treatment with C-PC reduced the level of the pro-apoptotic protein Bax by increasing the level of Bcl-2 in HEI-OC1 cells. Similarly, C-PC reduced the loss of mitochondrial membrane potential and increased ATP production in Madin-Darby Canine Kidney (MDCK) cells exposed to oxalate toxicity [21]. Lim et al. suggested that C-PC improved recovery from cisplatin-induced nephrotoxicity in mice and HK-2 cells by suppressing the expression of Bax, caspase-9, and caspase-3 [10]. Moreover, the previous study by Fernandez-Rojas et al. demonstrated the underlying mechanism that C-PC pretreatment attenuates alternations of calcium retention, ATP synthesis rate, and mitochondrial membrane potential sequentially following mitochondrial permeability transposition pore opening, which is occurred by the excessive ROS in the mitochondria isolated from kidneys of cisplatin-treated mice. This study also showed that the C-PC prevented the damage of renal function and structural abnormalities of mitochondria in the kidney of cisplatin-treated mice [22]. Our results, together with those of previous studies, suggest that the antioxidant activity of C-PC reduces cisplatin-induced toxicity. C-PC might, therefore, have clinical value as a nutraceutical to prevent or reduce hearing loss or renal injury in patients who receive platinum-based chemotherapy.

## 3. Materials and Methods

### 3.1. Culture Conditions of Limnothrix sp. KNUA002

We used a strain of *Limnothrix* sp. KNUA002 isolated from Lake Daecheong (36°22’N, 127°28’E) in September 2009. We inoculated the isolated cyanobacteria at a density of 5% (v/v) in BG-11 medium and cultured them for 10 days at 25 °C under a light intensity of 100 μmol m^−2^ s^−1^ with a 16 h/8 h light/dark cycle and shaking at 160 rpm. Cell cultures were maintained until they reached the log phase.

### 3.2. Acetate Buffer Extraction of C-PC from Limnothrix sp. KNUA002

We centrifuged a 100-mL culture at 3220 g for 15 min to obtain a cell pellet. We then suspended the pellet in 4 mL 20 mM acetate buffer containing 50 mM sodium chloride and 2 mM sodium azide [23]. To extract C-PC from the pellet, we subjected the pellet to four cycles of freezing at −20 °C, followed by thawing at room temperature (RT). We then centrifuged the extract at 3220 g for 15 min to remove cell debris. We determined the C-PC concentration from the optical densities at 615 nm and 652 nm using the following equation [24]: C-PC (mg/mL) = (A_615_ − 0.474(A_652_))/5.34.

### 3.3. Cell Culture and Viability Assay

We cultured HEI-OC1 mouse auditory cells under permissive conditions (33 °C, 10% CO_2_) for 16 h in Dulbecco’s modified Eagle’s medium (Hyclone, Logan, UT, USA) containing 10% fetal bovine serum (Hyclone) and 50 units/mL IFN-γ without antibiotics [25]. In all in vitro experiments, we seeded 7 × 10^3^ cells/well in 96-well plates, pre-treated the cells with 0.1, 0.5, 1, 2, 5, 10, or 20 μg/mL C-PC from *Limnothrix* sp. KNUA002 for 1 h, and treated the cells with 30 μM cisplatin for 30 h, unless otherwise indicated. We measured cell viability using 3-(4,5-dimethylthiazol-2-yl)-2,5-diphenyltetrazolium bromide (MTT; Sigma, St. Louis, MO, USA). After the cisplatin treatment, we added 0.5 mg/mL MTT solution to the culture media and incubated the cells for 2 h at 33 °C with 10% CO_2_. We measured the optical density of each well at 550 nm using a microplate reader.

### 3.4. Cell Cycle Analysis

After cisplatin treatment, HEI-OC1 cells were trypsinized, counted, centrifuged, and fixed in 70% ethanol. Cells were washed twice with phosphate-buffered saline (PBS) and centrifuged. We resuspended the centrifuged pellets in a solution of RNase A (0.2 mg/mL) and propidium iodide (10 μg/mL) and incubated them at 4 °C for 30 min. We determined the cell cycle distribution (10,000 events per sample) using a BD FACS Aria III flow cytometer (BD Biosciences, San Diego, CA, USA).

### 3.5. Detection of DNA Fragmentation

We analyzed apoptosis using an in situ cell death detection kit (Roche Biochemicals, Mannheim, Germany) based on the TUNEL technique. We cultured 3 × 10^4^ cells/well on 24-well culture plates and treated them with cisplatin for 30 h. We fixed the cells in 4% paraformaldehyde (PFA) for 15 min at RT, washed them, and permeabilized them for 5 min at 4 °C with freshly prepared 0.1% Triton X-100 and 0.1% sodium citrate in water. We then added TUNEL reaction mixture (50 μL) to the samples and incubated them for 60 min at 37 °C with protection from light. After washing the samples three times with PBS for 5 min each, we stained the nuclei with 4′, 6-diamidino-2-phenylindole (DAPI) for 5 min. We visualized the specimens using fluorescence microscopy (Carl Zeiss, Oberkochen, Germany).

### 3.6. Measurement of Intracellular ROS Production

We cultured HEI-OC1 cells for 16 h and then treated them with 30 μM cisplatin for 24 h with or without C-PC pretreatment. We measured intracellular ROS levels using the DCFH-DA (Invitrogen-Molecular Probes, Eugene, OR, USA). We washed the cells twice with 1 × PBS and incubated them with 10 μM DCFH-DA for 5 min at 33 °C and 10% CO_2_. We performed flow cytometry analysis (10,000 events per sample) using a BD FACS Aria III flow cytometer (BD Biosciences, San Diego, CA, USA).

### 3.7. Protein Preparation

We washed HEI-OC1 cells with ice-cold PBS and suspended them in 100 μL Radioimmunoprecipitation assay (RIPA) buffer (Elpis Biotech, Daejeon, Korea) containing Protease Inhibitor Cocktail Set 1 (Calbiochem, La Jolla, CA, USA). We transferred the suspension into a pre-cooled 1.5-mL tube, which we then incubated on ice for 15 min, centrifuged at 12,660 g for 30 min at 4 °C, and vortexed for 1 min. We aspirated the resulting supernatant and placed it in a fresh tube on ice, discarding the pellet.

### 3.8. Western Blot Analysis

We analyzed HEI-OC1 cell extracts by 12% sodium dodecyl sulfate–polyacrylamide gel electrophoresis. We electro-transferred the separated proteins to nitrocellulose membranes, blocked the membranes with a solution of 20 mM Tris-HCl (pH 7.6), 137 mM NaCl, and 0.01% Tween-20 (TBS-T) containing 5% skim milk for 1 h, and then probed the membranes with primary antibodies (1:100–1:2000 dilution) at RT. Next, we washed the membranes three times for 5 min each with TBS-T and incubated them with secondary antibody (1:2000 dilution). After a series of washes, we developed the membranes using an enhanced chemiluminescent detection system. The primary antibodies were rabbit anti-Bax and anti-caspase-3 antibodies from Cell Signaling (Danvers, MA, USA) and rabbit anti-Bcl-2 from Abcam (Cambridge, MA, UK). The secondary antibody was goat anti-rabbit-IgG-HRP (Cell Signaling).

## 4. Conclusions

In summary, we tested the ability of C-PC from *Limnothrix* sp. KNUA002 to protect auditory cells from cisplatin-induced ototoxicity. We present evidence that C-PC from *Limnothrix* sp. KNUA002 protects cells against the cytotoxicity induced by cisplatin. Pretreatment with C-PC inhibited apoptosis and protected mitochondrial function by preventing ROS accumulation in cisplatin-treated HEI-OC1 cells. C-PC from *Limnothrix* sp. KNUA002 protects cells against cisplatin-induced cytotoxicity by inhibiting the mitochondrial apoptotic pathway. The protection appears to be mediated by inhibition of the apoptotic pathway in mitochondria.

## Figures and Tables

**Figure 1 marinedrugs-17-00235-f001:**
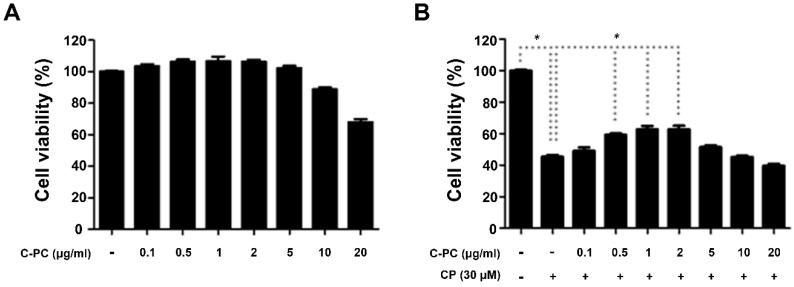
Effect of C-PC treatment on cell viability in cisplatin-treated HEI-OC1 cells. (**A**) Cells were cultured with 0.1, 0.5, 1, 2, 5, 10, or 20 μg/mL C-PC for 30 h. Cytotoxicity was evaluated by MTT assay. (**B**) Cells were treated with 0.1–20 μg/mL C-PC for 1 h and then treated with 30 μM CP for 30 h. Data represent the mean ± standard error of three separate experiments; * *p* < 0.05, compared with the cells treated with CP alone. C-PC, C-phycocyanin; CP, cisplatin.

**Figure 2 marinedrugs-17-00235-f002:**
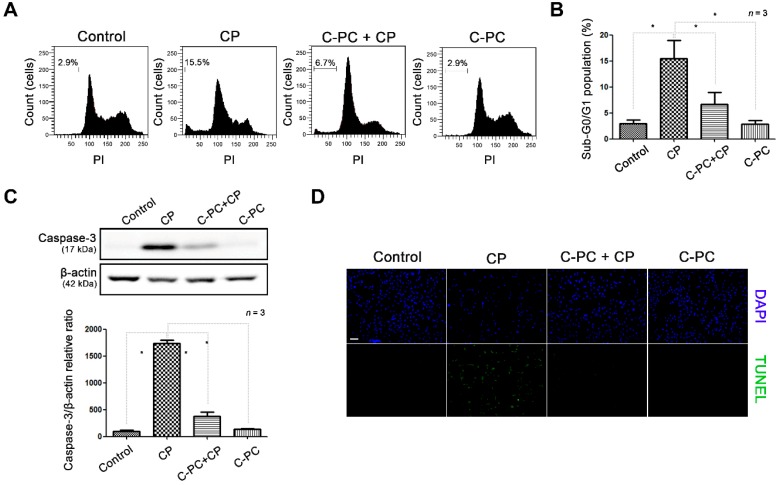
Effect of C-PC on cell cycle arrest and apoptosis in cisplatin-treated HEI-OC1 cells. (**A**) Cell cycle analysis by flow cytometry and (**B**) comparison of the sub-G0/G1 ratio between the cells treated with CP alone and those pretreated with C-PC. (**C**) Western blot showing caspase-3 expression in cells treated with CP and C-PC. Data are shown as the mean ± standard deviation; * *p* < 0.05, compared with the cells treated with CP alone. C-PC, C-phycocyanin; CP, cisplatin. (**D**) TUNEL assay to detect apoptotic cells. Fragmented DNA (green) and nuclei (blue) were stained and observed under fluorescence microscopy. Scale bar represents 100 µm. The cells were pretreated with 1 μg/mL C-PC for 1 h, followed by treatment with 30 μM CP for 30 h.

**Figure 3 marinedrugs-17-00235-f003:**
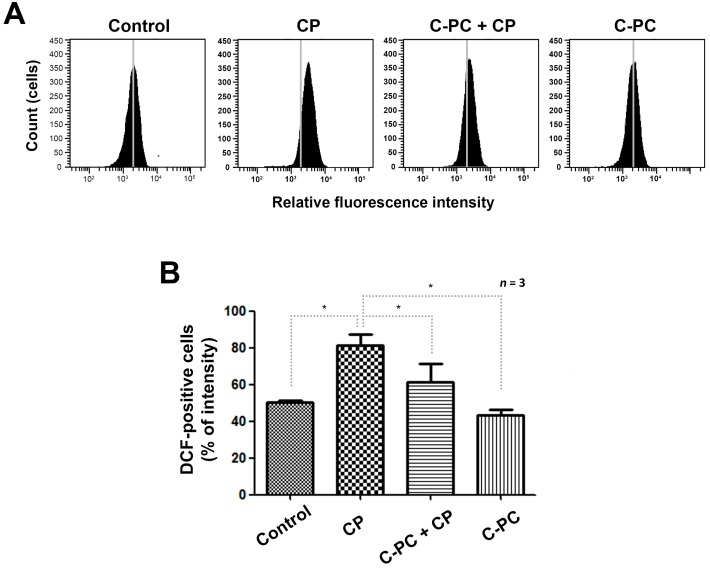
ROS scavenging capacity of C-PC in cisplatin-treated cells. (**A**) Measurement of intracellular ROS levels using a fluorescent dye 2′, 7′-dichlorodihydrofluorescein diacetate (DCFH-DA) probe. The fluorescence intensity was detected by flow cytometry. (**B**) Relative fluorescence-activated cell sorting (FACS) fluorescence intensities. The cells were cultured with 1 μg/mL C-PC for 1 h, followed by treatment with 30 μM CP for 24 h. C-PC, C-phycocyanin; CP, cisplatin. Data are shown as the mean ± standard deviation; * *p* < 0.05, compared with the cells treated with CP alone.

**Figure 4 marinedrugs-17-00235-f004:**
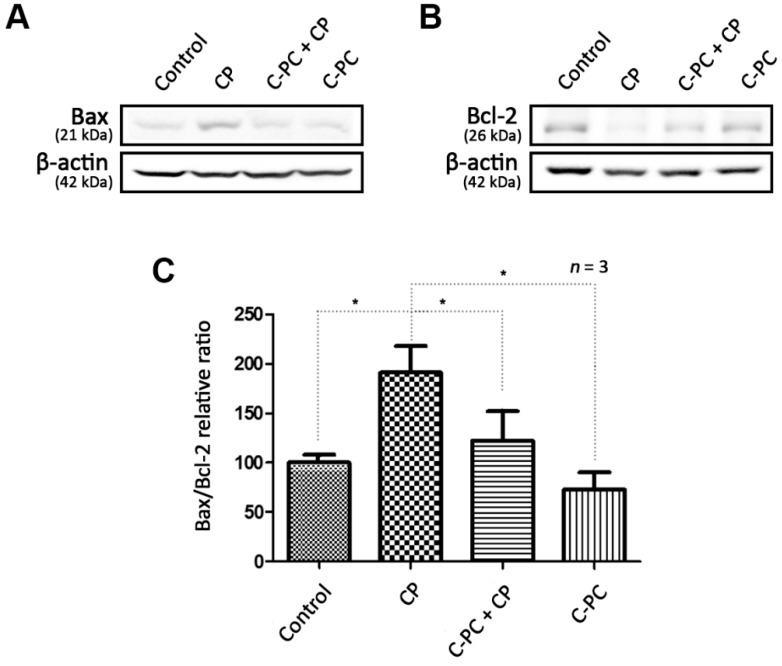
Effect of C-PC on Bax-mediated apoptosis caused by cisplatin. (**A**) Western blot showing Bax expression, compared with β-actin as the loading control (*n* = 3 per lane). (**B**) Western blot showing Bcl-2 expression, compared with β-actin as the loading control (*n* = 3 per lane). (**C**) Relative changes in the Bax/Bcl-2 expression ratio. CP treatment upregulated Bax expression compared with that in the control cells, and pretreatment with C-PC significantly alleviated the CP-induced increase in Bax expression. CP treatment downregulated Bcl-2 expression, and C-PC pretreatment alleviated the CP-induced downregulation of Bcl-2 expression. Cells were incubated with 1 μg/mL C-PC for 1 h and/or 30 μM CP for 24 h. Data are shown as the mean ± standard deviation; * *p* < 0.05, compared with the cells treated with CP alone. C-PC, C-phycocyanin; CP, cisplatin.

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
