# Peer review of "C-phycocyanin from Limnothrix Species KNUA002 Alleviates Cisplatin-Induced Ototoxicity by Blocking the Mitochondrial Apoptotic Pathway in Auditory Cells"

_marinedrugs, 2019, doi:10.3390/md17040235_

Round 1
Reviewer 1 Report
This is a well-designed study that presents novel findings using C-phycocyanin from Limnothrix species KNUA002 to prevent cisplatin induced damage to inner ear hair cells in vitro. The experiments are clearly described and the data support the hypothesis that C-PC from Limnotrix sp. 28 KNUA002 protects cells against cisplatin-induced cytotoxicity by inhibiting the mitochondrial apoptotic pathway in HEI-OC1 cells in vitro.
The results should lead to in vivo studies that would provide greater impact.
There are a few minor points that the authors should address.
What was the source for the HEI-OCI cells?
There are several typographical errors in the manuscript.
Line 106 “protean” should be “protein.”
Line 123 “intercellular” should be “intracellular.”
Figure 3 has misspellings of fluorescence and intensity.
Author Response
Response: HEI-OC1 (House Ear Institute-organ of Corti 1) cell is derived from the auditory organ of the transgenic mouse ImmortomouseTM, which harbor a temperature-sensitive mutant of the SV40 large T antigen gene under the control of an interferon- γ-inducible promoter element. It was first cloned and characterized by Kalinec et al. in 2003 [Kalinec GM, Webster P, Lim DJ, Kalinec F. A cochlear cell line as an in vitro system for drug ototoxicity screening. Audiol Neurootol. 2003Jul-Aug;8(4):177-89]. HEI-OC1 cells, which express several molecular markers including Myosin 7a, math 1, prestin, nestin as the characteristic of organ of Corti sensory cells, are sensitive to ototoxic drugs. Due to this unique property, it has been used as an excellent in vitro system to investigate the cellular mechanism(s) related to ototoxicity and to speculate the otoprotective properties of drugs or screening of the potential ototoxicity. We added this information in Materials and Method section as below.
Before ; We cultured HEI-OC1 mouse auditory cells under permissive conditions (33°C, 10% CO2) for 16 h in Dulbecco’s modified Eagle’s medium (Hyclone, Logan, UT, USA) containing 10% fetal bovine serum (Hyclone) and 50 units/mL IFN-γ without antibiotics.
After ; We cultured HEI-OC1 mouse auditory cells under permissive conditions (33°C, 10% CO2) for 16 h in Dulbecco’s modified Eagle’s medium (Hyclone, Logan, UT, USA) containing 10% fetal bovine serum (Hyclone) and 50 units/mL IFN-γ without antibiotics [14]
There are several typographical errors in the manuscript.
Line 106 “protean” should be “protein.”
Response : Protean is correct. As mentioned in the manuscript, we wanted to talk about the protean extract of spirulina platensis shows an increase in antioxidant [2].
Line 123 “intercellular” should be “intracellular.”
Figure 3 has misspellings of fluorescence and intensity.
Response : We changed all of typographical errors in manuscript.
References
1. Kalinec, G.M., P. Webster, D.J. Lim, and F. Kalinec, A cochlear cell line as an in vitro system for drug ototoxicity screening. Audiol Neurootol, 2003. 8(4): p. 177-89.
2. Bermejo-Bescos, P., E. Pinero-Estrada, and A.M. Villar del Fresno, Neuroprotection by Spirulina platensis protean extract and phycocyanin against iron-induced toxicity in SH-SY5Y neuroblastoma cells. Toxicol In Vitro, 2008. 22(6): p. 1496-502.
Reviewer 2 Report
The authors present the background that C-phycocyanin (C-PC), a blue phycobiliprotein found in cyanobacteria and red algae, has antioxidant and anticancer activities in different experimental models in vitro and in vivo. Thus, they tested the ability of C-PC from Limnothrix sp. KNUA002 to protect auditory cells from cisplatin-induced ototoxicity in vitro.
This is well described and prove to be successful.
As the authors mention the "mitochondrial apoptosis pathway" . They might want to discuss this more. As C-phycocyanin (C-PC) apparently is a tetrapyrrole, it might bind to TSPO and thereby exert its antioxidant and anti-apoptotic characteristics, as has been described previously for the mitochondrial protein TSPO.
Author Response
Response: As the reviewer’s suggestion, we added the underlying mechanism how C-PC prevents cisplatin-induced cytotoxicity from the sentences next to line 173 in the Results and Discussion section as following;
We added the sentences next to line 173 in the Results and Discussion section as following;
Lim et al. suggested that C-PC improved recovery from cisplatin-induced nephrotoxicity in mice and HK-2 cells by suppressing the expression of Bax, caspase-9, and caspase-3 [10]. Moreover, the previous study by Fernandez-Rojas et al. demonstrated the underlying mechanism that C-PC pretreatment attenuates alternations of calcium retention, ATP synthesis rate, and mitochondrial membrane potential sequentially following mitochondrial permeability transposition pore opening, which is occurred by the excessive ROS in the mitochondria isolated from kidney of cisplatin-treated mice. This study also showed that the C-PC prevented the damage of renal function and structural abnormalities of mitochondria in the kidney of cisplatin-treated mice [25]. Our results together with those of previous studies suggest that the antioxidant activity of C-PC reduces cisplatin-induced toxicity. C-PC might therefore have clinical value as a nutraceutical to prevent or reduce hearing loss or renal injury in patients who receive platinum-based chemotherapy.
Reviewer 3 Report
The current manuscript describes studies that demonstrated the effect of administration of C-PC from Limnothrix sp. KNUA002 on auditory cell line from cisplatin-induced ototoxicity in vitro. Pretreatment with C-PC from Limnothrix sp. KNUA002 inhibited apoptosis and protected mitochondrial function by preventing ROS accumulation in cisplatin-treated HEI-OC1 cells, a mouse auditory cell line. Tellingly, it was shown that cisplatin increased the expression of Bax and reduced the expression of Bcl-2, which activate and inhibit, respectively, the mitochondrial apoptotic pathway in response to oxidative stress.
Pretreatment with C-PC prior to cisplatin treatment caused non signficant changes in the Bax and Bcl-2 levels with respect to the levels in untreated control cells. These results suggest that C-PC from Limnotrix sp. KNUA002 protects cells against cisplatin-induced cytotoxicity by inhibiting the mitochondrial apoptotic pathway.
Overall, while the idea of this study might be potentially interesting, the current manuscript offers minimal describtional insights and it is based on just correlational aspects. Importantly, the current manuscript lacks any true novel molecular mechanistic insights.
I would give one example to clarify the lack of faithful and significant molecular insights in the current work. The author examined the abundance of some protein markers of cell death including BAX and BCl2 upon treatment cells with C-PC and found that there is alteration in the abundance of these important proteins upon C-PC treatment. A very important question would be how this effect occurs at least examining one potential mechanism. Does treatment impact post-translational modification of these proteins, does treatment impact the turnover or degradation of these proteins, or act through transcriptional means, does this treatment impact MOMP or any mitochondrial-related quality control mechanism like mitophagy? All these are very interesting issues that can be addressed or at least one or two of them can be addressed to give some significant molecular insights.
The major issue of this work is that this manuscript offers very minimal descriptional insights and not true molecular mechanisms.
Author Response
Response: We appreciate the helpful comments on the underlying mechanism how C-phycocyanin from Limnothrix species KNUA002 prevents cisplatin induced damage to inner ear hair cells in vitro. Cisplatin is one of the most widely-used drugs to treat various cancers, but its ototoxic and nephrotoxic side-effects remain major clinical limitations. Also, it has been known that cisplatin-induced ototoxicity and nephrotoxicity result from toxic levels of reactive oxygen species (ROS) followed by alteration of mitochondrial membrane potential and damage to the respiratory chain, which ultimately triggers the apoptotic process in cells. It is being considered that both ototoxicity and nephrotoxicity induced by cisplatin use similar intracellular mechanisms although more studies are needed to understand molecular mechanism of cisplatin ototoxicity. Under the potential consensus, the researchers have been studying for some common approaches to protect the cochlea and kidney from cisplatin toxicity, and antioxidants have been shown to reduce cisplatin-induced ototoxicity and nephrotoxicity. From this point of view, several studies have provided the mechanism underlying the protective effect of C-phycocyanin from cisplatin-induced toxicity. In particular, Fernández-Rojas et al. [2015] elaborately demonstrated the potential mechanism of C-phycocyanin pretreatment preventing cisplatin-induced mitochondrial dysfunction in mice [Fernández-Rojas B, Rodríguez-Rangel DS, Granados-Castro LF, Negrette-Guzmán M, León-Contreras JC, Hernández-Pando R, Molina-Jijón E, Reyes JL, Zazueta C, Pedraza-Chaverri J. C-phycocyanin prevents cisplatin-induced mitochondrial dysfunction and oxidative stress. Mol Cell Biochem. 2015 Aug;406(1-2):183-97]. The results of our study are also consistent with the previous studies that C-phycocyanin prevents the cisplatin-induced ototoxicity via mitochondrial dysfunction caused by reactive oxygen species leading to auditory cell death. In particular, the result of Bax/Bcl-2 protein expression may indicate that its alteration resulted from the loss of mitochondrial membrane potential following collapse of mitochondrial structure, and the treatment of C-phycocyanin attenuates mitochondrial abnormalities including MOMP. As the reviewer suggested, we addressed the molecular insight that the effect of C-phycocyanin resulted from protection of mitochondrial dysfunction in Discussion section as below since many studies support that both of cisplatin-induced cytotoxicity is closely related to increased ROS generation from which mitochondria is major source of ROS in the cell. We politely ask the reviewer’s understanding that we address its underlying mechanism in the Results and Discussion, and we will pursue the underlying mechanism in the next step of our research as the reviewer suggested.
We added the sentences next to line 173 in the Results and Discussion section as following;
Lim et al. suggested that C-PC improved recovery from cisplatin-induced nephrotoxicity in mice and HK-2 cells by suppressing the expression of Bax, caspase-9, and caspase-3 [10]. Moreover, the previous study by Fernandez-Rojas et al. demonstrated the underlying mechanism that C-PC pretreatment attenuates alternations of calcium retention, ATP synthesis rate, and mitochondrial membrane potential sequentially following mitochondrial permeability transposition pore opening, which is occurred by the excessive ROS in the mitochondria isolated from kidney of cisplatin-treated mice. This study also showed that the C-PC prevented the damage of renal function and structural abnormalities of mitochondria in the kidney of cisplatin-treated mice [25]. Our results together with those of previous studies suggest that the antioxidant activity of C-PC reduces cisplatin-induced toxicity. C-PC might therefore have clinical value as a nutraceutical to prevent or reduce hearing loss or renal injury in patients who receive platinum-based chemotherapy.